# The Fusion Gene *BPI-LY*, Encoding Human Bactericidal/Permeability-Increasing Protein Core Fragments and Lysozyme, Enhanced the Resistance of Transgenic Tomato Plants to Bacterial Wilt

**DOI:** 10.3390/plants14131897

**Published:** 2025-06-20

**Authors:** Lei Ni, Yue Zhang, Yafei Qin, Mei Wang, Daodao Tang, Liantian Chen, Xing Ding, Yilin Zheng, Yu Pan, Jinhua Li, Xingguo Zhang

**Affiliations:** 1College of Horticulture and Landscape Architecture, Southwest University, Chongqing 400715, China; nl1033148752@163.com (L.N.); zy18109037077@163.com (Y.Z.); qinyafei1995@163.com (Y.Q.); wangmei2817@163.com (M.W.); 15542164027@163.com (D.T.); 18223141946@163.com (L.C.); dingdingxing@email.swu.edu.cn (X.D.); zyil2021@163.com (Y.Z.); panyu1020@swu.edu.cn (Y.P.); ljh502@swu.edu.cn (J.L.); 2Key Laboratory of Agricultural Biosafety and Green Production of Upper Yangtze River (Ministry of Education), Southwest University, Chongqing 400715, China; 3Academy of Agricultural Sciences, Southwest University, Beibei, Chongqing 400715, China

**Keywords:** tomato, bacterial wilt, *Ralstonia solanacearum*, bactericidal/permeability increasing protein and lysozyme fusion gene (*BPI-LY*), disease resistance

## Abstract

Tomato bacterial wilt, caused by *Ralstonia solanacearum* (G^−^), is one of the most devastating plant diseases. Developing effective resistance against this pathogen remains a major challenge in plant disease management. In this study, we constructed a fusion gene *BPI-LY* by combining the gene encoding the lipophilic functional domains of human bactericidal/permeability-increasing protein (BPI) with the gene of human lysozyme (LY). The recombinant gene *BPI-LY* was heterologously expressed in yeast and tomato. Preliminary in vitro assays in yeast demonstrated that BPI enhances LY’s antibacterial activity against G^−^ bacteria. Furthermore, overexpression of *BPI-LY* in tomato delayed onset of the disease in the transgenic lines and lowered the degree of tissue damage and the number of bacteria present in the stems relative to those in the wild-type plant. Additionally, the expression levels of the *SlSOD*, *SlPOD*, *SlPAL*, *SlPR5*, *SlPR10*, and *SlPR-NP24* genes were indirectly upregulated in the transgenic plants following *R. solanacearum* inoculation. Collectively, these findings demonstrate that *BPI-LY* enhances the resistance of transgenic tomato against bacterial wilt caused by *R. solanacearum*.

## 1. Introduction

*Ralstonia solanacearum*, an aerobic Gram-negative (G^−^) bacterium, is one of the first plant pathogens for which the genome has been completely sequenced [1]. The pathogen infects plants through wounds on the roots and stems and cracks at the root tips or lateral roots and subsequently colonizes the root cortex, invades xylem vessels, and reaches the stem and aerial parts of the plant through the vascular system [2,3]. *R. solanacearum* reproduces rapidly in the xylem, forming a high density of bacterial colonies and producing extracellular polysaccharides, which block the plant vessels and affect water transport, causing the plant to wilt and eventually die [4]. *R. solanacearum* is the most destructive plant pathogen worldwide, with strains of this species comprising large populations that vary in geographic origins, host ranges, and pathogenic processes [5]. The host range of *R. solanacearum* includes 50 plants from over 200 families, including potatoes, tomatoes, tobacco, eggplants, and bananas [6]. After infection by pathogens, plants undergo a series of complex physiological and biochemical reactions, such as the production of reactive oxygen species (ROS), activation of protective enzyme systems, and regulation of disease-related genes.

Fleming [7] first discovered the lytic ability of lysozymes (LYs) in 1922. Human LY is naturally compatible with the human body and safer for clinical applications because it is non-irritating and has no side effects compared with other LYs [8]. In animals and humans, LYs serve as a protective barrier against bacterial pathogen colonization and invasion. However, although LY is widely present in the human body, such as in body fluids, cells, tissues, and organs [9], its extraction from humans is relatively rare and it is instead obtained through other means. In addition to their general antibacterial activity, LYs also exhibit antiviral activity. Soluble recombinant LY has already been successfully expressed in *Escherichia coli*, and its high biological activity has been demonstrated. It can also be obtained from yeast [10]. Nakajima et al. [11] found that after introducing the assembled *LY* gene into tobacco, the transgenic tobacco plants showed enhanced resistance to fungal and bacterial diseases.

The structure and synthesis of peptidoglycans are substantially different between Gram-positive (G^+^) and G^−^ bacteria [12]. Compared with that of G^+^ bacteria, the cell wall structure of G^−^ bacteria is more complex and the outer layer additionally contains lipopolysaccharide (LPS). LYs can degrade the peptidoglycan of the bacterial cell wall, thereby compromising the structural integrity of the cell wall and leading to bacterial lysis and death. However, in G^−^ bacteria, this process is hindered by the outer membrane containing LPS, which prevents LYs from accessing the inner layer of peptidoglycan. Consequently, LYs exhibit stronger bactericidal activity against G^+^ bacteria while demonstrating weaker effects on G^−^ species [13]. The shielding effect of the outer membrane in G^−^ bacteria can lead to these pathogens developing resistance mechanisms against LY. However, research has demonstrated that LYs can exhibit enhanced bactericidal activity against G^−^ bacteria following specific modifications [14].

Weiss et al. [15] were the first to isolate the bactericidal/permeability-increasing protein (BPI) from human neutrophils and discover its ability to kill G^−^ bacteria. BPI is an antimicrobial glycoprotein secreted by neutrophils and an endogenous cationic protein in humans and mammals. It is often found on the surface of eosinophils and monocytes as well as in mucosal and epithelial cells. Studies have shown that BPI exerts selective bactericidal activity against G^−^ bacteria and neutralizes endotoxins [16]. BPI has two domains: the N-terminal domain BPI1 and the C-terminal domain BPI2 [17]. The N-terminal domain binds to the LPS in the outer membrane of G^−^ bacteria [18]. BPIs from different species possess conserved amino acid sequences [19]. LPS triggers immune responses in various organisms, whereas BPI has bactericidal and LPS-binding activities [20]. A hybrid peptide composed of two regions of human BPI (amino acids 90–99 with bactericidal activity and amino acids 148–161 with LPS-neutralizing activity) has been shown to possess bactericidal capability against G^−^ bacteria [21]. In another study, a fragment of BPI was used to bind to the LPS of G^−^ bacteria to increase their membrane permeability and thereby lead to their destruction [22]. The LPS-binding protein/BPI-related proteins of *Arabidopsis thaliana* (*AtLBRs*) play a crucial role in inducing pathogenesis-related 1 (*PR1*) gene expression and ROS production in response to LPS [23]. Recombinant human BPI has been shown to have therapeutic effects against G^−^ bacterial infections [24]. Zhao et al. [25] found that a recombinant protein encoded by a gene obtained by fusing an artificially synthesized antimicrobial peptide-encoding gene with the *LY* gene exhibited inhibitory effects against *Staphylococcus aureus* (G^+^) and *E. coli* (G^−^). In recent years, the biological functions of BPI have been extensively investigated in disease resistance research [26].

Bacterial wilt of tomato is a devastating soilborne bacterial disease caused by *R. solanacearum* [27]. Infected plants exhibit wilting and drooping of the stems and leaves until they die completely. Traditional breeding techniques are time-consuming, and sources of disease-resistant varieties are limited. By contrast, genetic engineering technology can quickly and effectively screen resistant strains, thereby substantially improving breeding efficiency. This is an important measure for preventing and controlling bacterial wilt. However, the limitations caused by the widespread and diverse natures of pathogenic strains have resulted in a lack of efficient resistance genes in disease resistance research. To enhance the resistance of tomato plants to bacterial wilt, domestic and international research efforts have recently shifted toward strategies that utilize the overexpression of broad-spectrum antimicrobial substances that are either endogenous to tomato plants or from animal sources [28,29].

In recent years, the expression of recombinant *LY* DNA in *Pichia pastoris* has been successfully achieved, with the protein product showing good antibacterial activity [30,31]. In this present study, the DNA sequence encoding the functional lipophilic domain of BPI was fused with the human *LY* gene to form a *BPI-LY* fusion gene, in order to penetrate the outer wall of the hydrophobic lipopolysaccharide cell of G^−^ bacteria and destroy the inner wall of peptidoglycan, and synergistically interact with the functional domains of BPI to expand the antibacterial spectrum of LY and kill G^−^ bacteria such as *R. solanacearum*. Thus, we first expressed the recombinant protein BPI-LY in yeast GS115 and analyzed its antibacterial effect on G^+^ and G^−^ bacteria in vitro and whether it has an antibacterial broad-spectrum effect, especially investigating the antibacterial effect on G^−^. Subsequently, using genetic transformation experiments, *BPI-LY* was overexpressed in tomato to analyze its ability to convey plant resistance to bacterial wilt. This study will provide a new gene resource for the prevention and control of bacterial wilt disease in tomato and other crops.

## 2. Materials and Methods

### 2.1. Vectors, Bacterial Strains, and Preparation of Pathogenic Inoculum

The *E. coli* strain DH5α, the *Agrobacterium tumefaciens* strain LBA4404, and the *P. pastoris* strain GS115, as well as vectors pPIC9k, pVCT1412, and pVCT2346, were all preserved in our laboratory. LBA4440 was successfully activated in liquid Yeast Extract Broth (YEB) medium (Coolaber, Beijing, China) and exhibited growth on solid YEB medium at 28 °C. Similarly, GS115 was activated in liquid Yeast Extract Peptone Dextrose (YPD) medium (Coolaber, Beijing, China) and demonstrated growth on solid YPD medium at 30 °C. The *S. aureus* and *Bacillus subtilis* strains were provided by Professor Xiaobing Du (Southwest University, China), and their growth conditions were the same as DH5α, which activated in liquid Luria–Bertani (LB) medium (Coolaber, Beijing, China), and grew on solid LB medium at 37 °C. The tomato bacterial wilt strain GMI1000 was provided by Professor Wei Ding (Southwest University, China). The strain was activated to the logarithmic phase with B liquid medium (Coolaber, Beijing, China) at 28 °C (Tang et al. [32]). Once the concentration of the bacterial suspension reached 1 × 10^8^ CFU/mL, it was used for subsequent inoculation experiments.

The vendor information for all the reagents, kits, instruments, and software used in the in vivo test of plants in Section 2 is the same as that stated by Zhang et al. [33] and Qiu et al. [34]. The reagents, kits, instruments, and software used in the in vitro test of yeast in Section 2 were provided by Beijing Coolaber Biotechnology Co., Ltd. (Beijing, China, https://www.coolaber.com/ (accessed on 18 April 2025)) and Beyotime Biotechnology (Shanghai, China, https://www.beyotime.com/index.htm (accessed on 18 April 2025)).

### 2.2. Plant Materials and Inoculation Methods

The tomato material *Solanum lycopersicum* L. ‘Ailsa Craig’ (AC) was self-propagated in our laboratory. Tomato seeds were played in containment soil and vermiculite (3:1) in a nutrient bowl and placed in the climate of the incubator with a 16 h light/8 h dark cycle photoperiod, temperature of 25 °C, and humidity of approximately 60–70%. When tomato plants reached the four-leaf stage, the stems of the tomato were inoculated with 1 mL bacterial suspension of GMI1000, and the plants were subjected to follow-up disease resistance analysis experiments (Qiu et al. [34]).

### 2.3. Obtaining the Supernatant of Recombinant Protein Expressed in Yeast

*LY* and *BPI-LY* (gene *BPI*: CAA48684; gene *LY*: X75362) were amplified from the pVCT1412 vector carrying the target gene *sp-BPI-LY* using the PrimStar high-fidelity enzyme (Takara, Dalian, China), separately. The pPIC9k vector was subjected to double enzyme digestion with *EcoR* I and *Not* I (Takara, Dalian, China). Then, the obtained fragments *BPI-LY* and *LY* were ligated to the fragment pPIC9k at 50 °C for 30 min under the ligation of homologous recombinase to form the recombinant vectors pPIC9k-*LY* and pPIC9K-*BPI-LY*, respectively, with the target gene lying adjacent to the α-factor of the vector and a 6× His tag at the C-terminus. After Polymerase Chain Reaction (PCR) detection of the positive strains pPIC9k-*LY* and pPIC9K-*BPI-LY*, sequencing of their plasmids was performed by Beijing Tsingke Biotech Co., Ltd. (Beijing, China, https://www.tsingke.com.cn/ (accessed on 18 April 2025)). The plasmids were linearized via *Sac* I (Takara, Dalian, China) enzyme digestion, purified, and re-covered, and then used to transform competent cells of GS115 yeast. The cells were plated on SD-His-deficient medium (Coolaber, Beijing, China) and incubated at 30 °C until colonies appeared. The strains were selected and the ones carrying *LY* were labeled as GS115 (pPIC9k-*LY*) (hereinafter, LY), those with *BPI-LY* as GS115 (pPIC9K-*BPI-LY*) (hereinafter, BPI-LY), and the ones with the empty vector as GS115 (pPIC9k) (hereinafter, P). Protein expression was induced in Buffered Methanol-complex Medium (BMMY (Coolaber, Beijing, China)) containing methanol (final concentration of 0.5% (Coolaber, Beijing, China)) and Kanamycin (Kan, final concentration of 50 mg/L (Coolaber, Beijing, China)). The protein supernatant samples were taken every 24 h and the protein expression was verified by Western blot using mouse anti-His antibody (Beyotime, Shanghai, China) (Shi et al. [35], Appendix A). The supernatants with the highest expression levels were centrifuged and stored at −80 °C in a low-temperature freezer. The three types of protein supernatants were labeled as P (empty vector), LY, and BPI-LY, respectively. The gene sequences and primers used are shown in Appendix A.

### 2.4. Antimicrobial Activity Test Statistics

*E. coli* DH5α was grown to the logarithmic phase, after which the cells were collected via centrifugation, resuspended in PBS (Coolaber, Beijing, China), and diluted 100-fold. A 50 µL aliquot of the diluted cells was spread onto solid LB medium. After 15 min, four holes were created in the solid agar, and 30 µL each of PBS (negative control, CK–), P, LY, and BPI-LY protein supernatants were added to the respective holes. The plates were incubated overnight at 37 °C. After 18 h, the inhibition zone diameters were measured using a caliper, and photographs of the plates were taken. All experimental procedures were performed in a sterile environment in a laminar flow hood [36]. Each treatment group was set up in triplicate, and the experiment was repeated three times. The results were analyzed using GraphPad Prism software 9.4.1. The methods used to determine the anti-*A. tumefaciens* (LBA4404), anti-*B. subtilis*, and anti-*S. aureus* activities of the proteins were the same as described above.

### 2.5. Determination of the Bacterial Growth Curves and the Bactericidal Activities of the Antibacterial Proteins

Bacterial growth curves: As described in Liu [37], 50 µL each of LB medium (CK–) and P, LY, and BPI-LY protein supernatants were added to the respective *E. coli* cultures (3 parallel samples for each treatment group). The OD_600_ values of each treatment group were recorded at 0, 15, 30, 60, and 120 min using a microplate reader.

Bactericidal activities: As described by Liu [37], 50 µL each of 0.85% NaCl (CK– (Coolaber, Beijing, China)), P, LY, and BPI-LY protein supernatants were added to the respective *E. coli* cultures and the OD_600_ values were measured at 0, 15, 30, 60, and 120 min.

Each treatment was repeated three times. The experimental results were analyzed using GraphPad Prism software 9.4.1. The same method was used for *S. aureus*.

### 2.6. Determination of Conductivity

The increase in the conductivity of *E. coli* was measured using the method of Liu [37], with slight modifications. The following four groups were set up: the CK– control group (40 μL 0.85% NaCl + 50 μL bacterial solution + 5 mL 0.85% NaCl) and the treatment groups P (40 μL 0.85% P + 50 μL bacterial solution + 5 mL 0.85% NaCl), LY (40 μL 0.85% LY + 50 μL bacterial solution + 5 mL 0.85% NaCl), and BPI-LY (40 μL 0.85% BPI-LY + 50 μL bacterial solution + 5 mL 0.85% NaCl). Conductivity was determined at 0, 30, 60, and 120 min, with that at 0 min being represented by R_1_ and that at the other time points by R_2_. The 0.85% NaCl solution was treated in a boiling water bath for 5 min and its conductivity was represented by R_0_. The relative conductivity increase for each treatment was calculated as follows: (R_2_ − R_1_)/R_0_. Each treatment was set up in triplicate, and the experiment was repeated three times. The experimental results were analyzed using GraphPad Prism software 9.4.1. The same method was used to measure the conductivity of *S. aureus*.

### 2.7. Construction of Plant Expression Vector

The pVCT1412 vector carrying the target gene *sp-BPI-LY* and the pVCT2346 vector were subjected to double enzyme digestion with *Xba* I and *Xho* I (Takara, Dalian, China), respectively, resulting in 8229 and 663 bp fragments. These fragments were ligated overnight at 37 °C using T4 DNA ligase (Takara, Dalian, China) to form the recombinant vector pVCT2455-*sp-BPI-LY*. The “sp” secretion signal peptide carried on this vector ensures the extracellular expression of the *BPI-LY* gene [28]. The identification of positive strains and the preservation methods of strains are the same as described by Zhang et al. [33]. Structural diagrams of the vectors, gene sequences, and primers are shown in Appendix A.

### 2.8. Acquisition of Disease-Resistant Transgenic Tomato Lines

Disease-resistant transgenic tomato lines were obtained by genetically transforming the constructed plant overexpression vectors using the *Agrobacterium*-mediated method. Transgenic lines containing *sp-BPI-LY* were obtained using the tomato plant AC for transformation. For the T_0_ generation, the modified cetyltrimethylammonium bromide (CTAB (Coolaber, Beijing, China)) method was used to extract leaf DNA, and PCR was performed to detect positive transgenic lines. The reactions were carried out in the following conditions: 94 °C for 5 min, followed by 35 cycles of denaturation at 94 °C for 30 s, annealing for 30 s, extension at 72 °C for 1 min 10 s (calculated as 1 KB/min), and the cycle end at 72 °C for 10 min. Subsequently, RNA was extracted from the positive lines for quantitative real-time PCR analysis to determine the expression levels of the target genes in the positive plants. The reaction conditions were as follows: denaturation at 95 °C for 10 min, 40 cycles of 15 s at 95 °C, 30 s at the primer annealing temperature, 5 s at 65 °C, and data acquisition at 95 °C. Three high-expression lines were selected: BSP-2, BSP-3, and BSP-4. The specific operation steps of the experimental method here are the same as those described by Zhang et al. [33]. These three transgenic lines were further bred, and plants of the homozygous T_2_ generation were used for subsequent experiments. The experimental results were analyzed using GraphPad Prism software 9.4.1. The primers used are listed in Appendix A.

### 2.9. Determination of the Tomato Plant Disease Index

Using the WT as a control, 1 mL of *R. solanacearum* GMI1000 was injected into the base of the stems of the transgenic plants. The disease index of the inoculated tomato plants was statistically analyzed according to the bacterial wilt disease grading standards and disease index statistical method [32], and the plant characteristics were photographed and recorded on days 0, 7, and 11. Every 10 seedlings constituted a group, and the experiment was repeated three times.

### 2.10. Detection of the Number of R. solanacearum in Tomato Stems

After 11 days of pathogen inoculation, the stems of WT and transgenic plants were collected, surface disinfected with 75% ethanol, rinsed 3–5 times with sterile water, and then soaked in sterile water for 30 min. Subsequently, the bacteria-containing soaking liquid was mixed evenly and diluted to a gradient concentration, and an appropriate amount was spread evenly onto triphenyl tetrazolium chloride (TTC (Coolaber, Beijing, China)) solid medium. The plates were sealed and incubated upside down in a bacterial incubator at 28 °C. After 2–3 days, the number of *R. solanacearum* on the agar was counted [32]. Three biological replicates were set up for each treatment, and the experiment was repeated three times. The experimental results were analyzed using GraphPad Prism software 9.4.1.

### 2.11. Anatomical Observation of Tomato Stems

For the anatomical observation of tomato stem tissue, the base of the tomato stem was first inoculated with either sterile water (CK–) or *R. solanacearum*. On post-inoculation day 7, a 3–5 cm stem segment was cut from the bottom of the inoculation hole toward the morphological upper end using sterilized scissors. After longitudinal sectioning of this stem segment, the disease state of the tissue section was photographed and recorded. Three biological replicates were set up for this experiment. Additionally, using sterilized scissors, a 2–3 cm stem segment was cut from the top of the inoculation hole and immediately placed in a formaldehyde–acetic acid–alcohol (FAA (Coolaber, Beijing, China)) fixative for more than 48 h. These samples were sent to Servicebio (Wuhan, China, http://www.servicebio.cn (accessed on 18 April 2025)) for paraffin embedding, cross-sectioning, and staining. Three biological replicates were set up for this experiment.

### 2.12. DAB and NBT Staining of Tomato Leaves

The stems of the WT and transgenic plants were inoculated either with water (control groups) or *R. solanacearum* (treatment). Three days after inoculation, the leaves of the plants were subjected to overnight 3,3′-diaminobenzidine (DAB (Coolaber, Beijing, China)) and nitrotetrazolium blue chloride (NBT (Coolaber, Beijing, China)) staining [38] and photographed in 60% glycerol.

### 2.13. Quantitative Analysis of Gene Expression

Tomato leaves and stems inoculated with *R. solanacearum* at 0, 24, and 72 h post inoculation were collected, and RNA was extracted for quantitative real-time PCR analysis of the expression levels of *sp-BPI-LY*. Leaves were collected at 0 and 72 h to analyze the relative expression levels of the defense enzyme-related genes *SlSOD*, *SlPOD*, and *SlPAL*, and the pathogenesis-related genes *SlPR5*, *SlPR10*, and *SlPR-NP24*. The total RNA isolation (TRIzol reagent, Takara, Dalian, China) and extraction, cDNA synthesis (cDNA Synthesis Kit, Beyotime, Shanghai, China), and quantitative real-time fluorescence (SYBR Green qPCR Mix, Beyotime, Shanghai, China) assays used were based on the methods described by Zhang et al. [33].Three biological replicates were set with three technical replicates, and the experimental results were analyzed using GraphPad Prism software 9.4.1. Primer sequences are shown in Appendix A.

### 2.14. Transcriptome Analysis

Stems from the WT, BSP-2, BSP-3, and BSP-4 plants were collected at 0 and 3 days post-inoculation, placed on dry ice, and sent to Wuhan Metware Biotechnology Co., Ltd. (Wuhan, China, http://www.metware.cn (accessed on 18 April 2025)) for transcriptome analysis. Three biological replicates were set up for the experiment. Differentially expressed genes (DEGs) that were upregulated or downregulated by more than 2-fold in the WT and transgenic lines were selected, using the tomato gene *SlActin7* as a reference. *SlGA2ox2* (Solyc07g056670.2.1), *SlWRKY33* (Solyc09g014990.2.1), and *SlWRKY41* (Solyc01g095630.2.1) were selected as the upregulated genes and *SlMYB55* (Solyc10g044680.1.1) and *SlbZIP25* (Solyc03g046440.1.1) as the downregulated genes, and the transcriptome data were validated using quantitative real-time PCR [34]. The experimental results were analyzed using GraphPad Prism software 9.4.1. The primers used are listed in Appendix A.

### 2.15. Statistical Analysis

The GraphPad Prism version 9.4.1 software was used for statistical analyses and graphing. The data between two treatments were compared and statistically analyzed through the *t*-test. When the dependent variable across multiple treatments was a single variable, data comparisons and statistical analyses were performed using one-way ANOVA analysis of variance. When the analysis across multiple treatments involved two factors, data comparisons and statistical analyses were performed using two-way ANOVA analysis of variance.

## 3. Results

### 3.1. Recombinant BPI-LY Protein Increased the Bactericidal Activity Against G^−^ Bacteria

In the *in vitro* experiments, representative G^+^ and G^−^ bacteria were selected to evaluate the broad-spectrum antimicrobial activity of BPI-LY. Initially, recombinant proteins were expressed in yeast. Successful BPI-LY protein expression was demonstrated using Western blot analysis (Appendix A). Because the protein expression level was highest at 120 h, this time point was chosen as the optimal induction duration for subsequent experiments. The inhibition zone diameter of phosphate-buffered saline (PBS) in the culture media was 0 cm for all four types of bacteria (Figure 1a). A slight antimicrobial effect was observed with P, likely due to the presence of a low concentration of Kan antibiotic during the induction phase. However, the concentration of Kan used for inducing protein expression in all supernatants remained consistent. Compared with that of P, the inhibition zone diameter of LY was wider, but the inhibitory effect on *E. coli* DH5α and *A. tumefaciens* LBA4404 was not obvious. Compared with LY, BPI-LY had a substantial inhibitory effect on DH5α and LBA4404 (Figure 1a,b), indicating that BPI-LY enhanced the inhibitory effect of LY against the G^−^ bacteria. For the G^+^ bacteria, the inhibition zone diameters of LY and BPI-LY were substantially larger than those of P (Figure 1a). However, there was no notable difference between them in the strength of those effects (Figure 1b). The experimental results indicated that BPI-LY exhibited a superior antibacterial effect against G^−^ bacteria.

Studies have shown that the OD_600_ value is proportional to the number of bacteria and that the bacterial growth curve can be analyzed using absorbance measurements. Compared with the effects of CK–, the addition of the different protein supernatants resulted in a slower increase in OD_600_ over time, indicating a slower rate of DH5α growth. The antibacterial effects were BPI-LY > LY > P > CK–, with the addition of BPI-LY causing the growth curve of DH5α to become more stable, indicating that it had the best antibacterial effect (Figure 1c). The effects of the various proteins on *S. aureus* growth were similar (Figure 1d). As shown in Figure 1e, during 0–120 min, the OD_600_ of the DH5α culture treated with CK– changed linearly, indicating that it did not affect the growth of the strain. By contrast, after adding the protein supernatant, the OD_600_ gradually decreased, indicating a reduction in the viable cell count of DH5α. The trend in bacterial colony counts of the different groups was as follows: CK– > P > LY > BPI-LY. With regard to *S. aureus*, the number of viable cells in the groups treated with protein supernatant was lower than that of the control, with the trend in colony counts as follows: CK– > P > LY > BPI-LY. This result suggested that the antibacterial effects of LY and BPI-LY were short-term (Figure 1f). Consequently, BPI-LY demonstrated a pronounced inhibitory effect on G^−^ bacteria, yet exhibited a relatively weaker inhibitory effect on G^+^ bacteria.

Conductivity can be used to measure the permeability of cell membranes. As shown in Figure 1g, the conductivity in DH5α increased with time in each treatment group. Compared with those of the P and LY groups, the increase in conductivity of the BPI-LY group reached a substantial level at 60 min. At 120 min, BPI-LY had the most pronounced effect on DH5α cell membrane damage among all the tested compounds, with the highest conductivity increase reaching 0.19%. In *S. aureus*, the increase in conductivity in the LY and BPI-LY groups after 30 min reached a substantial level compared with that in the P group, indicating that LY and BPI-LY had begun to damage the cell membrane at this time point. Although the extent of damage increased with increasing time, the effects of the two proteins remained similar. At 120 min, the maximum increases in conductivity reached 0.0754% and 0.0794%, respectively, for the LY and BPI-LY groups (Figure 1h). The aforementioned experimental results demonstrated that BPI-LY increased the extent of damage to the cell membrane of G^−^ bacteria.

### 3.2. Creation of High-Expression sp-BPI-LY Transgenic Tomato Lines

*R. solanacearum* is a type of G^−^ bacteria. To analyze the antimicrobial function of *BPI-LY* directly in plants, we first constructed a plant overexpression vector. Sequence analysis revealed that the coding region of the modified *sp-BPI-LY* gene was 654 bp and encoded a 24.05 kDa protein composed of 217 amino acids. Among the amino acids, 30 were strongly basic, 13 were strongly acidic, 76 were hydrophobic, and 62 were polar. The primary structure of the pVCT2455-*sp-BPI-LY* vector is shown in Figure 2a. The plasmids from the vector-positive bacterial colonies were verified via restriction enzyme digestion, indicating the preliminary construction of the recombinant vector had been successfully achieved. Combined with the sequencing results returned by Wuhan Jinkairui Bioengineering Co., Ltd. (Wuhan, China, https://www.genecreate.cn/ (accessed on 18 April 2025)), the construction of the pVCT2455-*sp-BPI-LY* vector was verified to be correct.

In total, 69 rooted plants were obtained through *Agrobacterium*-mediated transformation (Figure 2b). PCR amplification revealed a fragment of approximately 1044 bp in 13 plants (Figure 2c), indicating that 13 *sp-BPI-LY* transgenic lines were successfully generated with a transgenic efficiency of 18.84%. Quantitative real-time PCR analysis was conducted on the T_0_ generation of the transgenic lines, using the transgenic line BSP-1 (with the lowest expression level) as the control. The relative expression levels in each transgenic line are shown in Figure 2d. Finally, the three lines with the highest expression levels (viz., BSP-2, BSP-3, and BSP-4 with expression levels 205.77, 174.56, and 176.26 times that of BSP-1, respectively) were selected for subsequent disease resistance tests.

### 3.3. Overexpression of sp-BPI-LY Enhanced the Resistance to Bacterial Wilt Disease in the Transgenic Lines

The statistics of disease index is helpful to analyze the disease resistance effect on plants. Potted WT (control) and transgenic tomato plants were inoculated with *R. solanacearum*. The WT plant began to exhibit wilting leaves after 2 days of inoculation, whereas the transgenic plants showed symptoms only on day 3. The tomato leaves became severely wilted over time, initially showing symptoms characteristic of bacterial wilt and then turning yellow and dry, with death of the entire plant ultimately ensuing. However, the time of disease onset occurred later in the transgenic lines, and the disease index was lower than that in the WT plant (Figure 3a). On day 11, the disease index of the WT plants reached 4.00, whereas the disease-resistant phenotype of the transgenic lines was highly apparent (Figure 3b and Appendix A). These results indicated that overexpression of the *sp-BPI-LY* gene in tomato plants enhanced their resistance to bacterial wilt.

### 3.4. Effect of sp-BPI-LY Overexpression on Disease Resistance in Transgenic Tomato Stems

*R. solanacearum* is a vascular pathogen, and assessing the extent of damage to tomato stems post-inoculation is critical. It is essential to determine whether the target gene is expressed in the stem. To this end, the expression levels of *sp-BPI-LY* in the tomato stem were analyzed, using the WT plant as the control (Figure 4a). The results showed that the *sp-BPI-LY* levels in BSP-2, BSP-3, and BSP-4 were substantially higher than those in the WT stem at 0, 24, and 72 h, confirming successful expression of the *sp-BPI-LY* gene in the stems of transgenic plants. The number of *R. solanacearum* in the stems of the high-expression transgenic lines was substantially lower than that in the WT stem (Figure 4b,c). On day 11, the amount of *R. solanacearum* in the WT stem was 3.65 × 10^9^ CFU/g, whereas the bacterial counts for BSP-2, BSP-3, and BSP-4 were 1.24 × 10^9^, 2.79 × 10^9^, and 1.91 × 10^9^, respectively (Figure 4c), showing statistically significant reductions. The degree of colonization of the bacterial wilt pathogen on the stem was consistent with the disease status of the tomato plants (Figure 3b). The results demonstrated that the expression of *sp-BPI-LY* effectively contributed to a reduction in bacterial content in the transgenic lines.

In addition, we performed anatomical observations on the inside of the tomato stem. After staining the paraffin-embedded transverse sections of tomato stems, we found that both WT and transgenic tomato tissues remained intact after water inoculation. However, after inoculation of the WT with *R. solanacearum*, the stain color of the vascular cylinder area became darker, with obvious holes in the pith (the place marked by the black arrow and number), indicating pathogen blockage, and notable damage to the pith tissue was evident. By contrast, there were no obvious holes in the pith of the transgenic plants, and the dark green part was reduced (the place marked by the black arrow and number), and the damage to the entire stem was significantly reduced, indicating that the degree of damage to the vascular bundles was substantially reduced in the *sp-BPI-LY*-expressing lines (Figure 4d). With regard to the longitudinal cross-sections of the tomato stems, no difference was found between the WT and transgenic lines after water inoculation. However, after inoculation with *R. solanacearum*, the cross-section of the WT showed severe disease symptoms, with a dark brown area near the inoculation site that gradually spread upward from the inoculation hole, accompanied by pus outflow. By contrast, the inoculation site of the transgenic tomato plants showed a lighter brown color with no pus outflow, and the stem color was still a healthy green color, indicating reduced pathogen invasion (Figure 4e), which indicated that *BPI-LY* had conveyed good disease resistance to the plant. In summary, these results suggested that the resistance of tomato stems to *R. solanacearum* could be enhanced by the expression of *sp-BPI-LY*.

### 3.5. Effect of sp-BPI-LY Overexpression on ROS Accumulation and the Expression of Defense-Enzyme-Related Genes and Pathogenesis-Related Genes in Transgenic Tomato Plants

The expression levels of the *sp-BPI-LY* gene in the leaves of the transgenic tomato plants were also measured, with the WT plant used as the control (Figure 5a). At 72 h after inoculation, the relative *sp-BPI-LY* expression levels of BSP-2, BSP-3, and BSP-4 were respectively 32,611.00, 12,900.21, and 38,714.27 times higher than that of the WT. These results confirmed the successful expression of the target gene in the leaves of the transgenic plants.

The production of ROS, including superoxide (O_2_^−^) and hydrogen peroxide (H_2_O_2_), represents one of the earliest cellular responses to pathogen infection. However, the over-accumulation of ROS may cause oxidative damage to plant cells. Leaf tissue sections were stained with DAB and NBT to detect the accumulation of H_2_O_2_ and O_2_^−^ within the *R. solanacearum*-infected plants (Figure 5a,b). According to the results, the leaves of the WT and transgenic plants showed no notable differences in H_2_O_2_ and O_2_^−^ accumulation after being injected with sterile water. However, after being infected with *R. solanacearum*, the WT leaves exhibited a deeper color than that shown by strain BSP-2. This suggested that the overexpression of *sp-BPI-LY* could reduce the accumulation of H_2_O_2_ and O_2_^−^ in tomato leaves during *R. solanacearum* infection.

The defense-related enzymes superoxide dismutase (SOD), peroxidase (POD), and phenylalanine ammonia-lyase (PAL) play an important role in plant resistance to abiotic stress. The relative expression levels of *SlSOD* in BSP-3 and BSP-4 reached highly substantial levels, being 23.40 and 21.83 times higher than that in the WT, respectively (Figure 5d). The relative expression level of *SlSOD* in BSP-2 also reached a highly substantial level at 72 h, being 5.62 times that of the WT (Figure 5d). At 0 h, the relative expression levels of *SlPOD* in BSP-3 and BSP-4 were substantially high, being 16.79 and 21.62 times that of the WT, respectively (Figure 5e). The relative expression levels of *SlPAL* in BSP-3 and BSP-4 were the highest, reaching substantially high levels that were 21.09 and 31.09 times that of the WT, respectively (Figure 5f). In summary, after inoculation with *R. solanacearum*, the genes responsible for synthesizing defense enzymes in transgenic lines were activated and expressed.

The expression of pathogenesis-related genes (*PRs*) is a typical feature of plants that activate their defense mechanisms. Compared with the WT, BSP-3 and BSP-4 showed substantially higher levels of *SlPR5* expression at 0 h, with BSP-3 showing the highest relative expression level (5.65 times that of the WT). At 72 h, the relative expression level of the gene in BSP-3 remained substantially high, reaching 3.31 times that of the WT (Figure 5g). BSP-3 and BSP-4 expressed substantial levels of *SlPR10* at 0 h, with BSP-3 showing the highest expression level (4.18 times that of the WT). At 72 h, the relative expression levels of *SlPR10* in BSP-2, BSP-3, and BSP-4 were also substantially high, with BSP-3 showing the highest level, reaching 3.50 times that of the WT (Figure 5h). *SlPR-NP24* expression in BSP-3 and BSP-4 reached substantially high levels at 0 h, with BSP-3 showing the highest level, reaching 2.75 times that of the WT. At 72 h, BSP-3 also showed the highest expression level of this gene, reaching 2.53 times that of the WT (Figure 5i). In summary, the expression levels of pathogenesis-related genes in the transgenic lines were generally higher than those in the WT plant, indicating that *sp-BPI-LY* overexpression promoted the expression of pathogenesis-related genes in tomato plants, enhancing their resistance to *R. solanacearum*.

### 3.6. Effect of sp-BPI-LY Overexpression on the Gene Expression of Tomato Infected with R. solanacearum

Transcriptome analysis serves as a powerful tool to enhance our understanding of the host’s disease resistance mechanisms. In total, the WT plant exhibited 5450 DEGs before and after *R. solanacearum* inoculation, with 2398 genes upregulated and 3052 downregulated. In the BSP transgenic lines, the number of DEGs before and after *R. solanacearum* inoculation was 3090, with 1477 genes upregulated and 1613 downregulated (Figure 6a,b). In total, the various lines had 2288 DEGs in common, among which five notable DEGs were involved in metabolic pathways and four were involved in secondary metabolic pathways. Among the upregulated genes shared between the transgenic lines and the WT, many WRKY family genes (e.g., *SlWRKY33*, *SlWRKY41*, etc.) related to disease resistance were found. The disease resistance-related protein gene *PR1a* was upregulated 2.35 times in the BSP lines. The gibberellin 2-dioxygenase 2 (*GA2ox2*) gene was the most highly upregulated, with its expression increased by 14.97 times in the WT plant and by 15.40 in the BSP lines (Appendix A).

Through Gene Ontology (GO) analysis, the DEGs were functionally annotated and found to be enriched in cell/cell part, catalytic activity, binding, metabolic, and cellular processes (Figure 6c). Kyoto Encyclopedia of Genes and Genomes (KEGG) analysis revealed the 14 most markedly enriched pathways of the DEGs (Figure 6d). After pathogen inoculation, the pathway most markedly enriched in *sp-BPI-LY* was the linoleic acid metabolism pathway, followed by the plant–pathogen interaction, phenylpropanoid biosynthesis, metabolism, biosynthesis of secondary metabolites, flavonoid biosynthesis, endocytosis, MAPK signaling—plant, and plant hormone signal transduction pathways. Pathogen inoculation substantially affected the gene regulation of many disease resistance-related pathways in the transgenic plants. The KEGG analysis results showed a series of physiological and biochemical changes occurring within the tomato plants after inoculation, revealing the metabolic background of resistance generated in response to *R. solanacearum* after overexpression of *sp-BPI-LY*.

Finally, we selected several genes from both the up- and downregulated gene sets identified in the transcriptome analysis and validated these findings using real-time PCR (Figure 6f,g). In the WT plant, the relative expression level of *SlWRKY33* was the highest, at 7.65 times that of the pre-inoculation level, whereas in the BSP transgenic lines, the relative expression level of *GA2ox2* was the highest, at 41.28 times that of the pre-inoculation level. The analysis demonstrated that the patterns of gene upregulation and downregulation were consistent across the dataset (Figure 6e–g), indicating that the latter results are reliable.

## 4. Discussion

### 4.1. Recombinant Protein BPI-LY Could Enhance the Antibacterial Activity Against G^−^ Bacteria

Compared with the cell wall of G^+^ bacteria, that of G^−^ bacteria is surrounded by an additional outer membrane LPS layer, which hinders the entry of LY. Studies have shown that fusion proteins can improve the in vitro antibacterial activity and spectrum of LYs [25,39]. The expression of recombinant proteins in *P. pastoris* was found to have a substantial antibacterial effect against both G^+^ and G^−^ bacteria [36,40]. Moreover, the BPI23-haFGF fusion protein, composed of the functional N-terminal rBPI-23 of BPI, has certain inhibitory and killing activities against G^−^ bacteria [41]. Mytichitin-A expressed by *P. pastoris* has antibacterial activity against *S. aureus* and *B. subtilis* but no marked inhibitory effect on G^−^ bacteria [42]. In this present study, fragments of the LPS domains from BPI were combined with LY to obtain the recombinant protein BPI-LY for expression in *P. pastoris* GS115. The agar well diffusion assay results showed that BPI-LY had a wider inhibition zone diameter against *E. coli* DH5α and *A. tumefaciens* LBA4404 compared with LY. Although the recombinant protein also exhibited marked inhibitory effects on *S. aureus* and *B. subtilis*, the antibacterial effect was not substantially different from that of LY (Figure 1a,b). The antibacterial activities of substances against G^+^ and G^−^ bacteria can be evaluated using a series of specific indicators. In this study, we used the P and LY proteins as controls. As determined from measurements at OD_600_, the growth rates of *E. coli* and *S. aureus* were the lowest in the presence of BPI-LY, indicating that the recombinant protein inhibited the growth of both bacterial species (Figure 1c,d). The results of the bactericidal tests showed that BPI-LY resulted in the lowest number of viable bacteria, demonstrating its bactericidal activity (Figure 1e,f). The conductivity test results indicated that BPI-LY had the greatest impact on the permeability of the *E. coli* cell membrane, showing stronger destructive effects than those of LY. However, there was little difference between BPI-LY and LY in their effects on *S. aureus* (Figure 1g,h). The conclusions of our experiments were consistent with the findings of previous studies [37]. In summary, BPI-LY had the same antibacterial activity against G^+^ as LY, but stronger antibacterial activity against G^−^ than LY, confirming that the recombination of LY and BPI can significantly enhance the antibacterial effect of LY against G^−^ bacteria (Figure 1a–h).

### 4.2. Overexpression of the sp-BPI-LY Gene Reduced ROS Damage in Transgenic Tomato and Enhanced the Expression of Defense Enzyme-Related Genes in Transgenic Tomato

When plants are attacked by pathogens, they rapidly produce ROS, including O_2_^−^ and H_2_O_2_ [43]. These molecules exhibit intrinsic antimicrobial activity, which can reduce pathogen viability, but they also exacerbate damage to the host plant [44]. The balance between ROS generation and detoxification determines the extent of cell death in infected plants [45]. In this study, we found that transgenic plants exhibited reduced ROS-induced damage (Figure 5b,c). Stress resistance in plants is related to their antioxidative capacity, which is generally achieved through the action of antioxidant enzymes. The activity of SOD, a key antioxidant enzyme, is the first step in the removal of ROS from plants. Additionally, POD activity has been shown to be positively correlated with disease resistance in plants [46]. PAL, a key enzyme in the flavonoid synthesis and phenylpropanoid metabolism pathways [47], is related to stress resistance in plants. The SOD gene family in tomatoes may undergo alterations following stress exposure [48], while in rapeseed, most members predominantly exhibit upregulation [49]. Enhancing stress tolerance can be achieved by modulating the changes in *SlPOD* and *SlPAL* [50]. In this study, at 0 h, the expression levels of *SlSOD*, *SlPOD*, and *SlPAL* were substantially higher than the levels in the control group, but gradually decreased over time. This was possibly due to an insufficient sampling time, leading to undetected peaks (Figure 5d–f). Our results suggested that overexpression of *sp-BPI-LY* improved bacterial wilt resistance by reducing ROS damage and indirectly increasing the expression of defense enzyme-related genes (Figure 5a–f).

### 4.3. Overexpression of the sp-BPI-LY Gene Enhanced the Expression of the Pathogenesis-Related Genes in Transgenic Tomato

When plants are attacked by pathogens and other biological stressors, they activate their defense mechanisms by inducing the expression of pathogenesis-related proteins (PRs) [51]. PR-5b may be involved in the plant response to biotic and abiotic stressors [52], and PR-5 proteins can degrade fungal cell walls and inhibit hyphal growth and spore germination [53]. The PR-NP24 protein was initially discovered in tomato and tobacco cell suspensions under NaCl stress [54]. PR-NP24 was found to convey resistance to fungal diseases in tomatoes [55]. PR-10 may be involved in defense mechanisms, and some of these proteins are induced and accumulated by viruses [56], fungi [57], and bacteria [58]. In this study, *SlPR-5*, *SlPR-10*, and *SlPR-NP24* expression was induced upon infection with *R. solanacearum*, reaffirming previous findings that PR proteins are indeed induced by bacterial diseases. Moreover, the *PR* gene expression levels of most of the high-expression transgenic lines were higher than those of the WT. The simultaneous high expression of *sp-BPI-LY* suggested that the overexpression of this exogenous gene enhanced the resistance of tomato to *R. solanacearum* by regulating the expression of *PR* genes, indirectly (Figure 5g–i).

### 4.4. Transcriptome Analysis of Tomato Overexpressing sp-BPI-LY

By determining the expression levels of specific genes, transcriptomics can be used to understand the molecular mechanisms underlying plant resistance to stress and disease [59]. A study discovered that DEGs in the highly resistant plants were related to the cell wall, endocytosis, and the genotype-specific regulation of defense-related genes after *R. solanacearum* infection [60], which is consistent with the results of this study (Figure 6c). Transcriptomic studies of tobacco have shown that its resistance to *R. solanacearum* is mainly related to the phenylpropanoid pathway. *NtPR1a* overexpression was found to increase the resistance of tobacco cv. Yunyan 87 to *R. solanacearum* by activating defense-related genes [61]. Transcriptomics showed that *PR1* was upregulated in tomato plants upon their inoculation with *R. solanacearum* [62]. After infection with *R. solanacearum*, many WRKY family genes (e.g., *WRKY34* and *WRKY41*) in peanuts exhibit upregulated expression, thereby enhancing disease resistance [63]. Similar to the conclusion above, in our study, *SlWRKY33*, *SlWRKY41*, and *PR-1a1* were significantly upregulated after inoculation with *R. solanacearum* (Figure 6e,f and Appendix A). Transcription factor families, such as bZIP and MYB, are closely associated with biological stress responses in plants. Among these genes, some function as positive regulators while others act as negative regulators [64,65]. In this study, the expression levels of *SlbZIP25* and *SlMYB55* were notably downregulated (Figure 6g, Appendix A). Transcriptome analysis showed that DEGs were mainly enriched in 11 pathways, including plant hormone signal transduction, biosynthesis of secondary metabolites, and plant–pathogen interaction [66]. We found that the differential gene functions caused by the pathogen-induced overexpression of *sp-BPI-LY* mainly manifested in 14 disease resistance-related pathways (Figure 6d), indicating the high functionality of this fusion gene in mediating plant disease resistance. During regulation of the biological response to *R. solanacearum*, the plant undergoes extremely complex reaction processes. Transcriptome sequencing can only provide a predictive analysis of the functions of disease resistance genes. The regulatory functions that they exert should be further verified through the expression of related products based on gene function predictions. Nevertheless, combining previous research findings with the conclusions of this study, it can be inferred that plant hormone signal transduction and metabolism-related pathways are important regulatory pathways for disease resistance genes (Figure 6c,d). These findings provide a research direction for elucidating the resistance mechanisms in tomato against bacterial wilt.

### 4.5. Biological Safety of sp-BPI-LY Transgenic Tomato

In China, genetically modified organisms such as soybeans, corn, rice, cotton, and papaya have been approved for commercialization. However, China adopts a cautious approach towards genetically modified technology and has established a series of stringent safety supervision systems. Genetically modified foods available on the market are clearly labeled to allow consumers to make free and informed choices. While genetically modified technology has brought significant convenience to people, its commercialization in China has been relatively slow. Compared with Western countries, there remains a noticeable gap in the theoretical research level of transgenic technology in China [67,68]. In our study, a recombinant *BPI-LY* gene was introduced into tomato plants, and the feasibility of this approach for enhancing resistance to bacterial wilt in tomatoes was evaluated. *BPI* and *LY* are both human genes (Appendix A), not allergens or toxic proteins. Currently, our work primarily focuses on functional validation; however, practical application requires further testing regarding the environmental safety and food safety of genetically modified crops, as well as consideration of consumer psychological acceptance. Our findings contribute additional theoretical possibilities to the study of resistance mechanisms against tomato bacterial wilt.

## 5. Conclusions

In this study, we engineered a fusion gene (*BPI-LY*) by combining the LPS-binding domains of BPI with LY. Functional validation through in vitro assays showed that BPI markedly enhanced LY’s antibacterial activity against G^−^ bacteria. Furthermore, in vivo assays revealed that the *BPI-LY* fusion gene significantly enhanced resistance against tomato bacterial wilt caused by *R. solanacearum* (G^−^). These findings provide novel genetic resources for combating this devastating disease and establish a high-resistance tomato germplasm through transgenic expression of *BPI-LY*. This study elucidates part of the resistance mechanism of the *BPI-LY* gene in combating tomato bacterial wilt; however, its detailed molecular mechanisms warrant further investigation and analysis.

## Figures and Tables

**Figure 1 plants-14-01897-f001:**
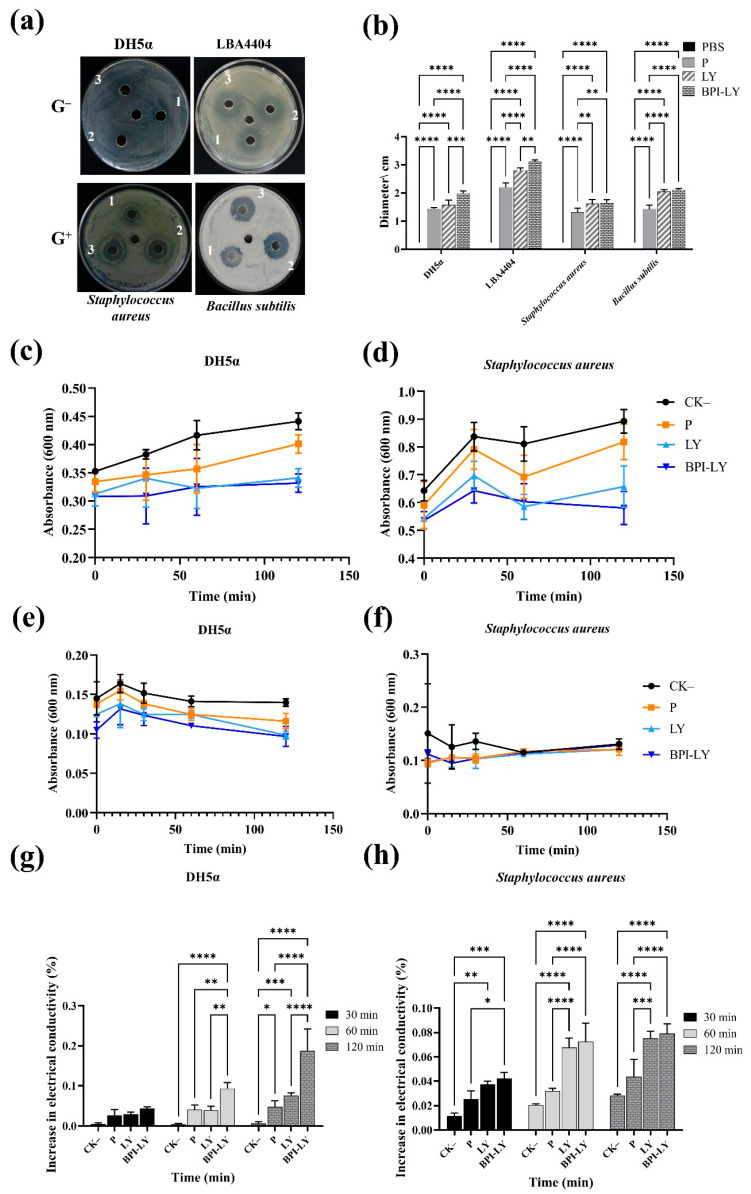
Antimicrobial activity of antimicrobial proteins *in vitro*. (**a**) Agar well diffusion test results. The middle hole represents PBS; 1: protein supernatant from empty vector P; 2: protein supernatant from LY; 3: protein supernatant from BPI-LY. The Gram-negative (G^−^) bacteria are *Escherichia coli* strain DH5α and *Agrobacterium tumefaciens* strain LBA4404. The Gram-positive (G^+^) bacteria are *Staphylococcus aureus* and *Bacillus subtilis*. (**b**) Antibacterial statistics of the agar plate. (**c**) Effect on the growth curve of DH5α. (**d**) Effect on the growth curve of *S. aureus*. (**e**) Effect on the bactericidal curve of DH5α. (**f**) Effect on the bactericidal curve of *S. aureus*. (**g**) Effect on the conductivity of DH5α. (**h**) Effect on the conductivity of *S. aureus*. All experiments were set up in triplicate and repeated three times. The results are expressed as the mean ± standard deviation, and statistical analysis was conducted using two-way ANOVA; α = 0.05, * *p* < 0.05, ** *p* < 0.01, *** *p* < 0.001, **** *p* < 0.0001.

**Figure 2 plants-14-01897-f002:**
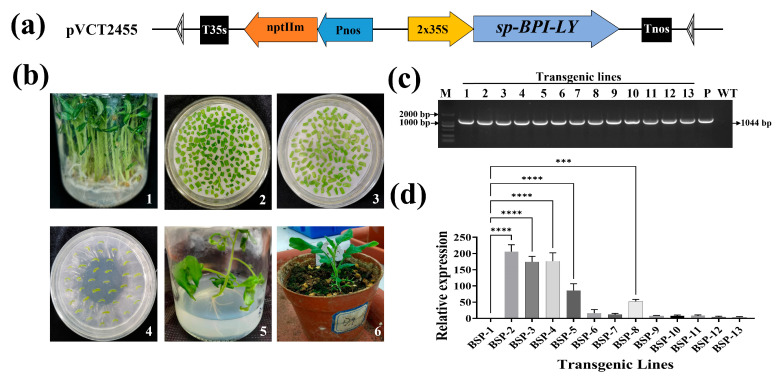
Acquisition of transgenic lines expressing the *sp-BPI-LY* gene. (**a**) T-DNA structure of the pVCT2455-*sp-BPI-LY* vector. T35s: Terminator of CaMV 35S gene; nptIIm: Neomycin phosphotransferase gene II; Pnos: Promoter of nopaline synthase gene; 2x35S: 2 x CaMV 35S Promoter; *sp-BPI-LY*: The fusion gene encoding secretion signal peptide (sp), the lipophilic functional domains of human bactericidal/permeability-increasing protein (BPI), and human lysozyme (LY); Tnos: Terminato of nopaline synthase gene. (**b**) Genetic transformation of tomato (1: aseptic seedlings; 2: precultured explants; 3: cocultured explants; 4: selected explants; 5: rooting cultivation; 6: transplanted seedling of transgenic plants). (**c**) PCR detection of the *sp-BPI-LY* gene in transgenic lines (M: 2000 bp DNA marker; 1–13: transgenic lines; P: plasmid positive control; WT: negative control). (**d**) Relative *sp-BPI-LY* expression levels of transgenic lines (BSP-1 to BSP-13 represent 13 transgenic lines positive for *sp*-*BPI-LY* expression). Three biological replicates were set with three technical replicates. The results are expressed as the mean ± standard deviation, and statistical analysis was conducted using one-way ANOVA; α = 0.05, *** *p* < 0.001, **** *p* < 0.0001.

**Figure 3 plants-14-01897-f003:**
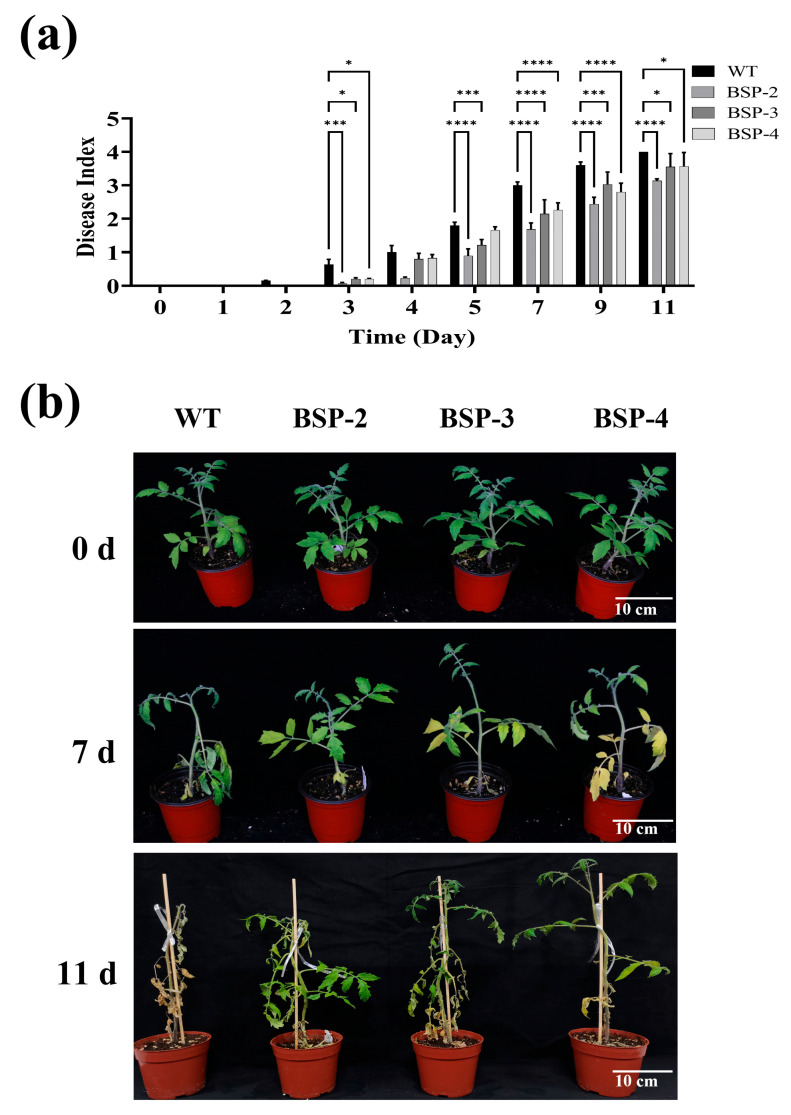
Phenotype and disease index of tomato after inoculation with *Ralstonia solanacearum*. (**a**) Disease indices for different growing days. Results are expressed as the mean ± standard deviation, and statistical analysis was performed using two-way ANOVA; α = 0.05, * *p* < 0.05, *** *p* < 0.001, **** *p* < 0.0001. (**b**) Comparison of seedling growth before and 7 and 11 days after inoculation; 0 d: Before inoculation; 7 d: inoculation for 7 days; 11 d: inoculation for 11 days. Note: WT: wild type; BSP-2, BSP-3, and BSP-4: T_2_ homozygous line offspring corresponding to the high *sp-BPI-LY* expression lines in the T_0_ generation. Every 10 seedlings constituted a group, and the experiment was repeated three times.

**Figure 4 plants-14-01897-f004:**
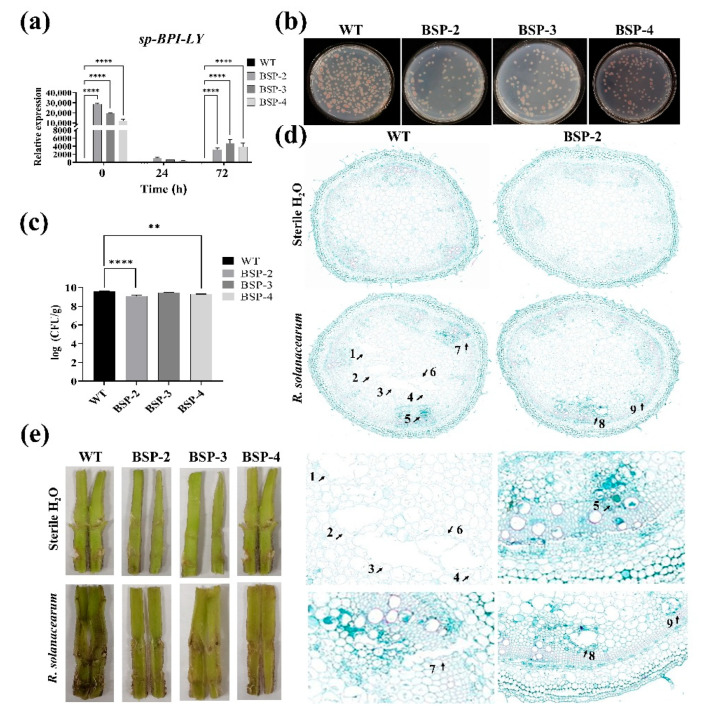
The effect of overexpression of *sp-BPI-LY* on tomato stems infected with *R. solanacearum*. (**a**) *sp-BPI-LY* expression levels in tomato stems of different transgenic lines at different times. Note: WT: wild type; BSP-2, BSP-3, and BSP-4: T_2_ homozygous line offspring corresponding to the high *sp-BPI-LY* expression lines in the T_0_ generation. Results are expressed as the mean ± standard deviation, and statistical analysis was performed using two-way ANOVA; α = 0.05, **** *p* < 0.0001. (**b**) Bacterial growth in the stems of WT and transgenic lines after 11 days of *R. solanacearum* infection. (**c**) Bacterial content in the stems of WT and transgenic lines after 11 days of *R. solanacearum* infection. Results are expressed as the mean ± standard deviation, and statistical analysis was performed using one-way ANOVA; α = 0.05, ** *p* < 0.01, **** *p* < 0.0001. (**d**) Paraffin-embedded cross-sections of stems. The first column shows the sterile water control, and the second column shows the inoculation effects. BSP-2 is the transgenic line with the highest *sp-BPI-LY* expression level. The small black arrow and number indicates the damage caused by *R. solanacearum*. 1–7: WT inoculated with *R. solanacearum*; 8–9: BSP-2 inoculated with *R. solanacearum*; the pictures above were the whole stem cross-sections, and the bottom pictures were the same sites after magnification. (**e**) Anatomical diagram of longitudinal sections of stems. The first column shows the sterile water control, and the second column shows the inoculation treatment effects. The experimental parts of a, b, and c used three biological replicates and were repeated three times, and the experimental parts of d and e used three biological replicates.

**Figure 5 plants-14-01897-f005:**
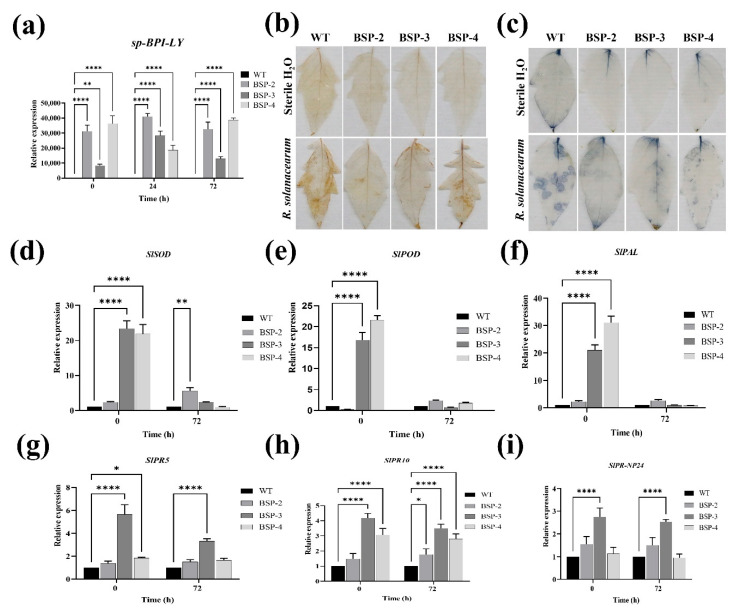
Effect of *sp-BPI-LY* overexpression on the ROS accumulation and the expression of defense enzyme-related genes and pathogenesis-related genes in tomato. (**a**) *sp-BPI-LY* gene expression levels in leaves of different transgenic tomato lines at different times. (**b**) DAB staining. WT represents the wild type; BSP-2, BSP-3, and BSP-4 represent the transgenic lines positive for *sp-BPI-LY* expression; sterile H_2_O is the control; and *R. solanacearum* is the disease causal agent; this is the same below. (**c**) NBT staining. (**d**) *SlSOD* gene expression. (**e**) *SlPOD* gene expression. (**f**) *SlPAL* gene expression. (**g**) *SlPR5* gene expression. (**h**) *SlPR10* gene expression. (**i**) *SlPR-NP24* gene expression. Results are expressed as the mean ± standard deviation, and the statistical analysis was performed using two-way ANOVA; α = 0.05, * *p* < 0.05, ** *p* < 0.01, **** *p* < 0.0001. The experimental parts of (**b**,**c**) used three biological replicates, and the experimental parts of (**a**,**d**–**i**) used three biological replicates and were repeated three times.

**Figure 6 plants-14-01897-f006:**
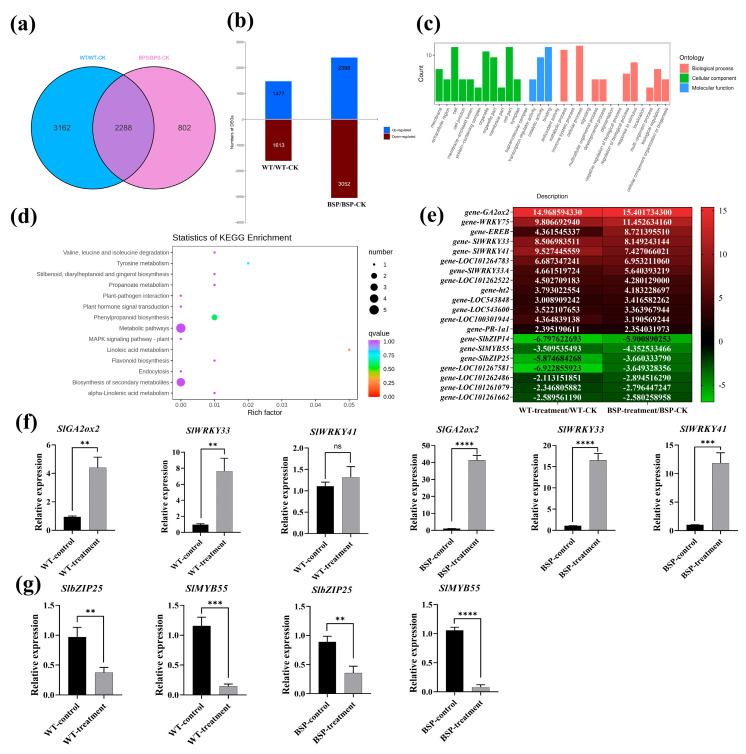
Transcriptome analysis results and real-time PCR verification of differentially expressed genes (DEGs). (**a**) Venn diagram of DEGs in the WT and transgenic lines. WT represents the wild type after pathogen inoculation, WT-CK represents the wild type before inoculation, BSP represents the *sp-BPI-LY*-positive transgenic strain after inoculation, and BSP-CK represents the transgenic strain before inoculation; this is the same below. The non-overlapping regions on the Venn diagram represent differential genes unique to the group, whereas the overlapping regions represent DEGs common to the several groups. (**b**) Number of DEGs between the WT and transgenic lines; up represents upregulated DEGs, whereas down represents downregulated DEGs. (**c**) GO classification map of DEGs in the *sp-BPI-LY* (BSP) lines after pathogen inoculation, with WT as a control. (**d**) Statistics of KEGG enrichment of DEGs in the *sp-BPI-LY* (BSP) lines after inoculation, with WT as a control. The ordinate represents the KEGG pathway, and the abscissa represents the Rich factor, which is the ratio of the number of DEGs annotated to an item to the total number of genes annotated to that item. The larger the Rich factor, the greater the degree of enrichment. The size of the dots represents the number of genes enriched in this pathway: the larger the number, the bigger the dots. The colors of the dots represent the significance of enrichment for this pathway, assessed with the corrected *P*. The closer the *p*-value is to 0, the redder the color is, which represents a more substantial enrichment. (**e**) Heat map of some DEGs in transcriptome data. The ordinate represents the differential gene, and the abscissa represents the different treatments. The color ranges from green to red, with a redder color representing a higher amount of expression. (**f**) Validation of upregulated genes in the WT and BSP transgenic lines. ns: not significant. (**g**) Validation of downregulated genes in the WT and BSP transgenic lines. All samples were collected from three biological replicates of each treatment at specified intervals. The results are expressed as the mean ± standard deviation, and statistical analysis was performed using *t*-tests; α = 0.05, ** *p* < 0.01, *** *p* < 0.001, **** *p* < 0.0001.

## Data Availability

The data that support the results are included in this article and its Appendix A.

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
