# Peer review of "The Fusion Gene BPI-LY, Encoding Human Bactericidal/Permeability-Increasing Protein Core Fragments and Lysozyme, Enhanced the Resistance of Transgenic Tomato Plants to Bacterial Wilt"

_plants, 2025, doi:10.3390/plants14131897_

Round 1
Reviewer 1 Report
Comments and Suggestions for Authors
In the study, the authors engineered a fusion gene (BPI-LY) by combining the LPS-binding domain of BPI with LY. Functional validation through in vitro and in vivo assays revealed that the BPI-LY fusion gene significantly enhanced resistance against R. solanacearum, the causative agent of tomato bacterial wilt. These findings are very meaningful which provide novel genetic resources for combating this devastating disease and establish a high-resistance tomato germplasm through transgenic expression of BPI-LY.
But there are some points needs to be revised or corrected, such as:
P8, in Fig 1, section g, 60min, the significance of difference between CK- and P is ns may wrong;
P12,in Fig 4, The fonts in the picture are too small, and in the part d, the small black arrow indicating the damage caused by R. solanacearum are not clear;
P15, Fig 6, part f and g are annotated incorrectly in the title, they are Inconsistency between the caption and the image; and the expression levels of these test genes should be compared with the transcriptome data and q-PCR results.
Reviewer 2 Report
Comments and Suggestions for Authors
In the current manuscript, the authors attempted to examine whether the BPI-LY fusion gene can confer resistance to bacterial wilt disease in tomato plants. Overall, the study is well designed; however, some of the results appear to be at borderline in backing the claims. The following are the major concerns that should be addressed before it can be published.
The result section should introduce a sentence explaining the objectives of the particular experiments and include a conclusion statement at the end of each section. At many places, the experimental design details are lacking. Figure legends should describe figures appropriately. Sentences appear incomplete and non-informative.
Result 3.1: Was BPI-LY protein created in this study? If so, please include the details of fusion protein creation. Was it the culture supernatant added in the holes to see the zone of clearance? Why does the empty vector show a zone of clearance? Many of the differences are non-significant.
Result S1: Please explain the western blot. Which tag was used to create the recombinant protein? It is not clear which antibodies were used to detect both the BPI-LY and the positive control in the same western blot.
Figure 1 legends – Antibacterial statistics of the agar plate mean nothing. Please write the description of the figure along with the number of biological replicates. Correct this in all the figure legends.
Result section 3.2 – The result title does not read well. Change the title to “Creation of high-expression BPI-LY transgenic tomato lines”.
Result 3.2: What was the transformation efficiency? Figure legend 2c: PCR of transgenic lines; what does it mean? Figure legend does not match the figure. There are just 13 samples not 22, no CK+ or CK-. What was amplified in the PCR? Please include a schematic diagram of PCR validation design.
The title of the result 3.3 is vague. What is meant by ‘obvious’? Please use scientific language while describing the findings of the study.
Result 3.3a: There are four bars at each time point, however, the figure displays just 3 keys.
Figure 4a-c : Panels are too small to evaluate.
Comments on the Quality of English Language
It can be improved.
Reviewer 3 Report
Comments and Suggestions for Authors
The manuscript "Expression of the Fusion Gene BPI-LY in Tomato Plants 2
Conveys Resistance to Bacterial Wilt" describes an original and promising methodology to generate resistance to plant pathogen Ralstonia solanacearum in the tomato crop. However the structure should be reorganized for a better reading of the text. Results should be presented in agreement to Materials and Methods, or vice versa. Also, the order of Tables and Figures in Supplementary material must be in agreement to their mention in the text. Statistical analyses, which are extensive, must be justified with evaluation of errors' normality distribution of the variables under study. In Discussion, the fact that there are antecedent of resistance to this plant pathogen by means of incorporating plant genetic resources must be considered, especially taking into account than transgenic crop for fresh market are difficult to introduce in the production systems because of negative public perception. In this case, this negative public perception is potentially increased because of the introduction of a human gene in tomato plants. By considering these suggestions, the manuscript will increase its quality of presentation and scientific soundness, and in consequence, a higher overall merit will be achieved.
Reviewer 4 Report
Comments and Suggestions for Authors
The introduction is well structured and it describes the reason of the work, its aims and the topics covered. The Material and Methods need to be better detailed in some experiments. A paragraph on statistical analysis is missing. The Results also need to be corrected in some parts. The results are well discussed and conclusions well supported. All issues and suggestions are presented as highlighted words and sticky notes, in the attached pdf file, named " plants-3624694-peer-review-v1 reviewer revised".

Reviewer 5 Report
Comments and Suggestions for Authors
In this manuscript, the authors attend to use the antibacterial fusion proteins from human to generate a transgenic tomato plant to control bacterial wilt disease. Based on this object, the manuscript is worth to discuss agriculture application. However, the manuscript still has logical arguments that need to be clarified and explained:
- Why were many microorganisms irrelevant to the subject of this manuscript used in the antibacterial experiments, but no antagonism assay performed with bacterial wilt pathogen (the most important pathogen in this paper)?
- The data presented in the chart is not consistent with the description in the results.
- Every experiment must be carried out with a basis or idea, which is lacking in this current manuscript. It is recommended to add an introductory sentence before each Materials and Methods.
The following are suggestions based on each paragraph:
Abstract
L32 increased in the transgenic under…(conditions or treatments)
L67 Please describe the function of lysozyme before describe the structure of cell wall
L84 A hybrid peptide composed of two regions of human … If possible, please describe something about will the structure change occur from LY, BPI to LY-BPI? The structure would change the function of hybrid proteins.
L109-116 In this paragraph, the authors may describe a flow on how to demonstrate the ideas through the experimental design.
Materials and Methods
Many methods just describe the source by citation, however, brief methods or procedures is still necessary.
L133-137 Growth conditions of plants and pathogens is required.
L139 “spBPI-LY”? where is the “sp” come from?
L142 “…by Zhang et al. [31].” What is the goal of subsequent procedures?
L201 delete “group”
L215 please provide the version of the software.
L224 please confirm GS115-pPIC9K-BPI is correct.
L244 “anti-B. subtilis, and anti-S. aureus.” Why R. solanacearum was not included in the assay since this pathogen is the core in this study?
However, I don’t think the antibacterial effect against B. subtilis, A. tumefaciens. E. coli, or S. aereus is necessary to put in this manuscript.
Please add the statistical analysis in the last paragraph in materials and methods.
Results
L270-279 I don’t understand why the experimental design was focused on these four types of bacteria, the main issue of this MS is bacterial wilt.
L298-299 “The suggestion …” is suitable to put in the discussion. Same as L359-360, L379-380, L393-395, L416-418, L422-424, and L445-448.
L351 The transgenic lines used in this study were homozygous or not, please explain.
L356 not “disease severity” If the authors want to calculate the control efficacy, the disease severity may necessary, not the disease index.
L374 If the population of R. solanacearum in all types of plants were over 10^9 CFU/g plant, the expression of figure 4b would have some problem.
L376-377 When presented with the disease index, it seems that the statistical differences are somewhat difficult to determine between transgenic lines and wild type plant. However, without a reduction in disease, subsequent experiments or discussions would seem to be meaningless. Please confirm this point again.
L391 the resolution of figure 4d and 4e is not easy to observe the differences. Please explain the results more detail here.
L419 The exogenous gene? Means spBPI-LY?
L423-433 How can we confirm that the instability shown in various data of different transgenic lines is derived from the trans genes?
L445 Please confirm what are the “disease-related genes”.
L460 If the acquisition of disease resistance in plants is a fact, how can we convince readers that the induced disease resistance comes from a protein with strong bactericidal activity?
Round 2
Reviewer 4 Report
Comments and Suggestions for Authors
Almost all issues and suggestions have been addressed by authors. Therefore, the manuscript can be accepted after these little corrections:
L122: ...solanacearum; Then we first... replace the semicolon with a dot or use lowercase letters for then.
L152: placed
L226-238: after revision you still did not explain the PCR conditions, but only referred to Zhang et al.
L301: .... analyzed through one-way or two-way ANOVA. ...
L367-369: all in italics.
L439 and 442: .... pith (The .... the with lowercase.
L526: a and b or a, b and c? And c-i or d-i?
Reviewer 5 Report
Comments and Suggestions for Authors
I observed that the author has responded to and revised most of the questions, but there are still some that need to be answered or written in the article.
- Please revised all of the leading sentences in the “results”, just describe the basic concept the authors attend to do without adding references. For example, in Line 367, please revised as “ solanacearum is a type of G– bacteria. To analyze the antimicrobial function of BPI-LY directly in plants, we first constructed a plant overexpression vector.” Then, put “The “sp” secretion signal peptide carried on this vector ensures the extracellular expression of the BPI-LY gene [38].” In materials and methods 2.7
- In the original version described “On day 11, the number of colonies 372 on the WT stem was 3.65 × 109 CFU/g, whereas the colony counts for BSP-2, BSP-3, and 373 BSP-4 were 1.24 × 109, 2.79 × 109, and 1.91 × 109, respectively (Figure 4b)”. However, these results were disappeared in the revised version. I think this data should be retained in revised manuscript. In my opinion, although the bacterial (colony) count in each plant reached more than 10^9, it is fine if there is no significant difference between them, because this unsignificant might become the connection that the author wants to continue to analyze whether the induced disease resistance was occurred. It is recommended to use the modified figure and write the original data in the results, and then confirm it with statistical analysis. At the same time, include this idea in the discussion.
- If the author has conducted an antibacterial activity analysis on Ralstonia solanacearum, it is still recommended to include it.
